# Biological vs. Crystallographic Protein Interfaces: An Overview of Computational Approaches for Their Classification

**Katarina Elez** [1,†] **, Alexandre M. J. J. Bonvin** [1,*] **and Anna Vangone** [2,*]

[1] Bijvoet Center for Biomolecular Research, Faculty of Science – Chemistry, Utrecht University, Padualaan 8, 3584 CH Utrecht, The Netherlands; katarina.elez1@gmail.com

[2] Pharma Research and Early Development, Large Molecule Research, Roche Innovation Center Munich, Nonnenwald 2, 82377 Penzberg, Germany

[*] Correspondence: a.m.j.j.bonvin@uu.nl (A.M.J.J.B.); anna.vangone@roche.com (A.V.)

[†] Present address: Department of Mathematics and Computer Science, Freie Universität Berlin, Arnimallee 6, 14195 Berlin, Germany.

**Abstract:** Complexes between proteins are at the basis of almost every process in cells. Their study, from a structural perspective, has a pivotal role in understanding biological functions and, importantly, in drug development. X-ray crystallography represents the broadest source for the experimental structural characterization of protein-protein complexes. Correctly identifying the biologically relevant interface from the crystallographic ones is, however, not trivial and can be prone to errors. Over the past two decades, computational methodologies have been developed to study the differences of those interfaces and automatically classify them as biological or crystallographic. Overall, protein-protein interfaces show differences in terms of composition, energetics and evolutionary conservation between biological and crystallographic ones. Based on those observations, a number of computational methods have been developed for this classification problem, which can be grouped into three main categories: Energy-, empirical knowledge- and machine learning-based approaches. In this review, we give a comprehensive overview of the training datasets and methods so far implemented, providing useful links and a brief description of each method.

**Keywords:** protein-protein interface; biological interface; crystallographic interface; classification; webserver; X-ray crystallography; protein structure; machine learning

## 1. Introduction

Proteins are considered the building blocks of cells. By interacting with each other and with other biomolecules, they control and elicit almost every biological process. Their function is highly regulated by the type and nature of the interactions they form, and any alteration in these networks can lead to disease [1]. Consequently, the study of protein interactions at the structural level plays a crucial role for the investigation of biological systems and for drug development.

For several decades, considerable efforts have been devoted towards the determination of three-dimensional (3D) structures of protein-protein complexes. The vast majority of high-resolution structures has been obtained by X-ray crystallography, representing the 89% of all the protein structures deposited in the Protein Data Bank (PDB, https://www.rcsb.org), as of October 2019 (Figure 1).

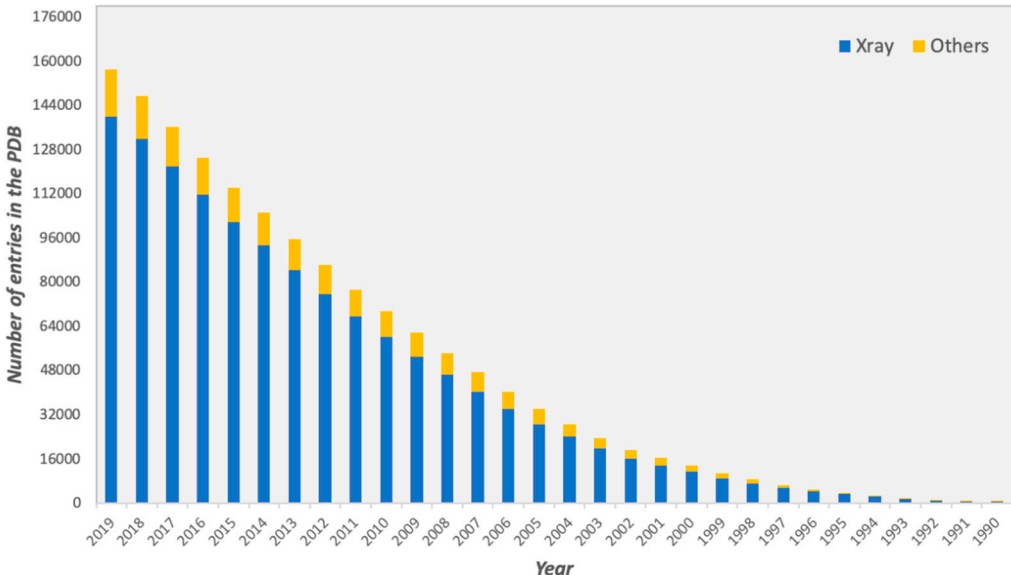

**Figure 1.** Total number of structures in the Protein Data Bank (PDB) from 1990 to October 2019. Structures determined by X-ray crystallography are depicted in blue, while structures determined by all other experimental techniques, such as Nuclear magnetic resonance (NMR) and Electron Microscopy or a combination of different methods, are shown in yellow.

Despite being the most commonly used experimental technique, X-ray crystallography presents two major challenges. The first is technical and related to the difficulty of growing high-quality protein crystals of sufficient size and form to enable structure determination [2]. The second is related to the analysis and interpretation of the resulting structures. At a molecular level, a crystal is formed by an infinite repetition of unit cells, which can result in various interfaces being formed (Figure 2): The biologically relevant one, corresponding to the interface occurring in solution and eliciting the biological function; and crystallographic ones, which are mere artefacts of crystal packing. Specifically, assemblies having a $K_d$ value in the low micromolar range or smaller are usually considered biological, unlike weak interactions which are characterized by higher $K_d$ values [3,4]. Further information regarding the extensive topic of protein interactions can be found in dedicated reviews, such as those by Nooren and Thornton [5] or Marsh and Teichmann [6].

Due to the complexity of the biological assemblies studied nowadays, the classification of biological vs. crystallographic interfaces with only crystallographic data is not always trivial and can be prone to errors. The increasing number of experimentally determined 3D structures of protein complexes over the years has allowed for a deeper analysis of the physico-chemical characteristics of biological versus crystallographic interfaces, helping the researchers to identify some key features that help to distinguish them. Overall, biological interfaces are rather specific in terms of amino acid composition, undergo evolutionary pressure and are thermodynamically more stable. In contrast, crystallographic interfaces are usually considered non-specific and are governed mostly by kinetically driven associations [8]. Based on those studies, a number of computational methodologies have been developed in order to distinguish those interfaces and are now playing a major role in protein interface classification.

In this review, we give a comprehensive overview of computational methodologies proposed so far to classify protein interfaces as biological or crystallographic ones. Brief descriptions of methodologies, performances reported by the authors and eventual availability (i.e., user-friendly web interface and/or source code) are given. Finally, a brief overview of the available datasets to train and validate prediction methods is also reported. For other valuable reviews in the field, please refer to those developed by Capitani et al. [4] and Xu et al. [9].

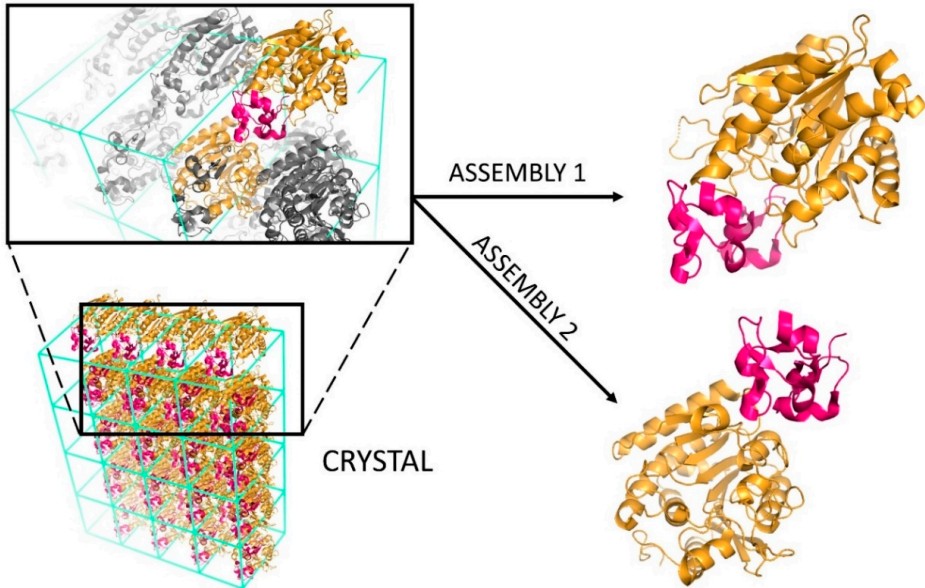

**Figure 2.** Schematic representation of the biological interface classification. In the lower left part of the figure, the crystal of the heterodimer between HLA class I histocompatibility antigen (in yellow) and beta-2-microglobulin (in dark pink) is reported (PDB code: 1ZLH [7]). Each unit cell of the crystal lattice contains two heterodimers. According to the architecture of this crystal, HLA can share two different interfaces with the beta-2-microglobulin: One formed with beta-2-microglobulin belonging to the same unit cell (assembly 1) and another formed with beta-2-microglobulin belonging to the adjacent unit cell (assembly 2).

## 2. Computational Methodologies for Classification of Biological Interfaces

Biological and crystal interfaces differ in several features. Parameters such as size of the interface, estimated solvation energy, number of salt bridges, knowledge-based atomic pair-wise potential and evolutionary conservation have been used traditionally to distinguish between the two types of interfaces. What has been clear for a while is that no single protein property on its own has been shown to be specific enough to distinguish the two types of interfaces. Highly refined description or combination of such features is necessary to develop accurate classifiers.

In the following, we report the most common methods so far described in the literature. Those are listed in Table 1 in alphabetic order. They are grouped in main categories according to the methodology implemented: Energy-, empirical knowledge- and machine learning-based methods. Energy-based techniques determine the stability of an interface using a scoring function which approximates the energy of a complex. The methods in the other two groups consider different properties of the interface, either by deriving heuristic rules (empirical knowledge-based), or by using specific algorithms (machine learning-based). When a combination of different approaches was adopted, we classified the approach based on the main features used. Finally, approaches that rely on a comparison between interfaces of homologous proteins across multiple crystal structures also exist, such as ProtCID [10] and QSbio [11], but they were not the focus of this work.

For each method, we report the prediction performance as originally reported by the authors. The performance metrics mentioned throughout the review are defined as:

$$\text{sensitivity} = \frac{\text{TP}}{\text{TP} + \text{FN}} \tag{1}$$

$$\text{specificity} = \frac{\text{TN}}{\text{TN} + \text{FP}} \tag{2}$$

$$\text{precision} = \frac{\text{TP}}{\text{TP} + \text{FP}} \tag{3}$$

$$recall = \frac{TP}{TP + FN} \tag{4}$$

$$accuracy = \frac{TP + TN}{TP + FN + TN + FP} \tag{5}$$

where TP, FN, TN and FP are the number of true positives, false negatives, true negatives and false positives, respectively. In this context, the positive class is defined as biological, while the negative class is that of the crystallographic interfaces.

## 2.1. Energy Based Classification Approaches

### 2.1.1. ClusPro-DC (2017)

ClusPro-DC [12] is based on the traditional docking algorithm ClusPro [13], whose protocol has recently been extended to perform interface classification. First, it separates the subunits of the dimer, docks them many times without any restrictions and retains the top 1000 poses with the lowest energy. Then, the number of near-native structures is determined by counting the docked poses whose C$\alpha$ interface RMSD is less than 7 Å from the crystal structure. If a reasonably large number of near-native states can be observed, then the method concludes that the interaction is stable and, therefore, is likely to be biologically relevant. Conversely, the dimer is considered unlikely to be biological if there are not many docked poses close to the crystallographic structure. Specifically, a threshold of 33 was chosen as optimal on the Bahadur et al. dataset [14,15] (Table 2). The classification accuracies of PISA, EPPIC and ClusPro-DC on the DC dataset [16] were reported by the ClusPro-DC authors as being 59.6%, 78% and 74.5%, respectively. On an automatically compiled dataset of 293 multimers and 490 monomers, the method slightly outperforms both PISA and EPPIC. When a subset of this dataset made of entries for which PISA and EPPIC contradict each other is considered, ClusPro-DC shows high performance, suggesting that this method can complement the information provided by the other two classifiers.

### 2.1.2. PISA (2005, 2007)

PISA (Protein Interfaces, Surfaces and Assemblies) [17,18] is currently the de facto standard for biological interface classification. It first performs a graph-theoretical search in order to find all possible assemblies in a crystal, ignoring in the process those containing identical parallel chains (leading to infinite size). The stability of the assemblies is then analyzed by calculating the free energy change upon dissociation from estimates of the free energy of binding and of the entropy of complex dissociation. Finally, in order to identify the correct biological unit, all stable assemblies are ranked using the following preferences (in order of priority): Larger assemblies, single-assembly sets (isolated subunit) and assemblies with higher free energy of dissociation. The method's four empirical parameters were optimized on the Ponstingl et al. 2003 dataset [19] (Table 2), resulting in an overall accuracy of 90%. The most frequent misclassification of PISA involved tetramers being classified as dimers, in accordance with a previous study based on the same dataset [19]. In 2015, an updated version of the webserver, called jsPISA [20], was released (Table 1). It is characterized by a modern user interface and two novel features: Assembly stock and interaction radar. The former helps in determining the oligomeric state of the protein under varying experimental conditions, while the latter is a radar plot showing the probabilities of a specific interface being biological.

## 2.2. Empirical, Knowledge-Based

### 2.2.1. Liu et al. (2014)

Liu and colleagues [21] investigated whether the B-factor, a measure of atomic vibrational motion, is more relevant for the classification of biological interfaces than the widely used interface area. They exploit four B-factor related features: B-factor score (sum of normalized B-factors of all interface

atoms), average B-factor score, number of interface atoms with a negative normalized B-factor and a combination of the last two. For all the newly introduced features, they find that the cross-dataset classification performance (by training the method on one dataset and validating on other three, repeating the process four times) is better than the one showed by the interface area. When the method is trained on the DC dataset [16] (Table 2) the average B-factor score is more accurate than both PISA [18] and EPPIC [16] on the Zhu et al. [22], Ponstingl et al. [23] and Bahadur et al. [14,15] datasets. This single feature also shows much higher specificity values and was demonstrated to be useful for correcting annotation errors.

### 2.2.2. EPPIC (2012, 2014, 2018)

EPPIC (Evolutionary Protein-Protein Interface Classifier) [16] relies on the number of core interface residues as the only geometric criterion and on two other evolutionary criteria: Core to rim and core to surface entropy ratios. Core residues are defined as those having more than 95% of their accessible surface area buried upon interface formation [24], a definition that has proven more effective than some previous formulations [25,26]. The selection pressure at the interface is estimated by retrieving homolog sequences with an optimal sequence identity cutoff of 60% (extended down to 50% if less than 10 homologs are available) and comparing the sequence entropies of core to rim and core to surface residues. A majority voting scheme is applied in all but the following cases: If an interface is larger than 2200 $\text{Å}^2$, then it is automatically classified as biological; when a sufficient number of homologs is not available, only the geometry criterion is used; if only one of the evolutionary scores and the number of core residues can be calculated, the preference is given to the former. The method was trained on the DC dataset [16] (Table 2), reaching an accuracy of 81%. On the Ponstingl et al. 2003 dataset [19] used for training PISA, they reached an accuracy of 89% vs. 84% for PISA. Furthermore, in a subsequent study, the authors compiled the more extensive Many dataset [27] (Table 2), for which they report an accuracy of 88%.

EPPIC is available as source code and as a user-friendly online webservice. The most recent version of their webserver, EPPIC 3 [28], switches to a reduced amino acid alphabet for sequence entropy calculation and outputs a probabilistic score for each interface based on a logistic regression classifier trained on the Many dataset [27]. The probability of an assembly is a combination of the pairwise interface scores and can be interpreted as a confidence value. On a dataset of 1481 PDB entries [28], EPPIC 3 and PISA show a roughly identical precision, with the former under-predicting and the latter over-predicting large assemblies.

### 2.2.3. PreBI and COMP (2006, 2008)

PreBI [29] discriminates homodimer interfaces from crystal contacts by considering the complementarity and the area of the interface. It automatically generates all possible interfaces according to the symmetry operators provided for each crystal structure and represents the molecular surfaces using vertices. Next, it calculates hydrophobicity, electrostatic potential and shape (curvature) for each vertex. The complementarity of an interface is estimated as the number of complementary pairs of vertices: pairs less than 1 Å apart having the same signs for the hydrophobicity, the opposite signs for the electrostatic potential and the opposite signs for the curvature. A simple sum of normalized individual complementarities and the area of the interface are combined through a series of heuristic rules, obtained by analyzing a self-assembled dataset of 393 and 344 homodimer and crystallographic interfaces, respectively. A filtered version of the dataset was subsequently used for training COMP [30], a linear combination of the same three complementarities. Hydrophobicity and shape were found to be important for the classification, while electrostatic potential did not show significant differences between the two classes. COMP successfully identified 84.8% of biological interfaces in the training dataset, which was brought to 88.8% after manual inspection.

### 2.2.4. CFPScore (2006)

CFPSscore (Combinatorial Four-Parameter Score) is a methodology based on a combination of four parameters: Potential mean force score (i.e., a statistical potential developed to estimate binding free energy of protein-protein interactions), interface size, packing density and shape complementarity [31]. Different combinations of cutoffs for those parameters were explored on a set of both biological and crystal interfaces (Postingl dataset [23]). The authors reported a prediction accuracy of 96.6%. They also showed the potential of their methodology for the problem of selecting native structures in docking decoy sets.

### 2.2.5. Bahadur et al. (2004)

In their work, Bahadur and colleagues compared interfaces in protein-protein complexes and homodimers to crystal-packing contacts made by monomers [15]. In addition to their previously compiled dataset of 122 homodimers [14], they assembled a new one consisting of 188 monomers that form crystallographic interfaces of at least 800 $\text{Å}^2$ (Table 2). They analyzed the following properties: interface area, polar/non-polar composition and interactions, buried interface atoms and core residues, amino acid composition, shape, atomic packing density, residue propensity and hydrophobic interaction energy. With respect to crystal artifacts, biological interfaces were found to be larger, more hydrophobic, to contain more fully buried atoms and to have a higher percentage of core residues. They were also attributed a better shape complementarity, better packing and a higher hydrophobic interaction score. The authors showed that a simple product between the non-polar interface area and the fraction of buried atoms could correctly identify 88% of homodimers and 77% of monomers in the training set. When the residue propensity score was taken into consideration as well, the success rates (sensitivities, see Equation (1)) rose to 93% and 95%, respectively.

### 2.2.6. Elcock and McCammon (2001)

Elcock and McCammon [32] hypothesized that residues located at biological interfaces are more conserved than the surface ones located in the non-interacting regions. In order to estimate the degree of conservation, they calculate a modified sequence entropy over 6 residue groups, instead of the usual 20 residues, for each position in the sequence. This mean interface entropy is then compared to the mean *surface* entropy and their ratio is used as the only classification criterion on the dataset by Ponstingl et al. [23]. The method correctly assigned nearly all examples that are wrongly classified by PITA but showed an almost identical accuracy as PITA. The authors point out an important disadvantage of the proposed criterion: Since it considers all surface residues, it has problems calculating the real entropy ratio for proteins with multiple binding sites for different partners on their surface.

### 2.2.7. PITA (2000, 2003)

One of the first methods for distinguishing between homodimers and monomers in crystal structures used the buried surface area and a simple score based on atom-pair frequencies across the interface [23]. Two atoms from different protein chains are counted as a pair if they are less than 8 Å apart. A distance dependent score is then obtained by considering the pair frequencies between atoms belonging to 17 different types according to the connectivity with their covalent partners. The method was trained and tested on their own dataset comprising of 96 monomers and 76 homodimers (Table 2). This approach was subsequently extended with a hierarchical graph partitioning procedure for automatic determination of quaternary structure and implemented in a webserver called PITA (Protein InTerfaces and Assemblies) [19]. A new dataset, Ponstingl et al. [19] (Table 2), containing 55 monomers and 163 oligomers, was compiled for the purpose of training and testing the predictor. Its overall accuracy (see Equation (5)) using atom-pair scoring was 84%, which is a marginally better performance with respect to the one obtained when using only the interface area as prediction feature (with a cutoff

of 1000 Å$^2$: all interfaces larger than the cutoff defined as biological, otherwise crystallographic). The major pitfall of the method was in differentiating tetramers from dimers.

*2.3. Machine Learning-Based*

2.3.1. PIACO (2019)

Covariation signals, extensively used in the field of protein contact prediction, were recently exploited in the PIACO (Protein Interface Analysis using COvarying signals) [33] predictor for an accurate classification of biological interfaces. The method uses Protein Sparse Inverse COVariance (PSICOV) [34] in order to estimate the covariation signal from large multiple sequence alignments. Specifically, the number of contact pairs having a PSICOV score higher than given thresholds (0.1, 0.2, 0.4 and 0.6) is used, together with other known features such as number of core residues, amino acid composition of the interface, amino acid composition of core residues, amino acid pair frequency, local density index, residue propensity score and gap volume index. Covariation signal features were found to be among the most important and along with 32 selected known features were used for training a random forest classifier on the DC [16] dataset (Table 2). In a 5-fold cross validation, the method achieves an accuracy (see Equation (5)) of 85%, compared to 74% and 80% obtained by PRODIGY-CRYSTAL and EPPIC (trained on this dataset), respectively. It also outperforms the other two predictors in terms of accuracy (see Equation (5)) on the Bahadur et al. [14,15] and Zhu et al. [22] datasets, with PRODIGY-CRYSTAL showing the highest sensitivity (see Equation (1)) and EPPIC showing the highest specificity (see Equation (2)) among the three predictors on all three datasets. Due to its evolutionary nature, PIACO is only applicable to entries that pass PSICOV's diversity criterion (84–90% of the datasets). When evaluated on the Many [27] dataset, it obtains an accuracy of 91%, while PRODIGY-CRYSTAL and EPPIC reach 92% (in a 10-fold cross validation) and 89%, respectively. The accuracy of PIACO improves further when trained on this dataset, rising to 95% in a 10-fold cross validation procedure.

2.3.2. PRODIGY-CRYSTAL (2018, 2019)

PRODIGY-CRYSTAL combines intermolecular residue contacts together with properties of the non-interacting surface and interaction energies (electrostatic, Van der Waals and desolvation) [35] for distinguishing between crystallographic and biologically relevant interfaces in protein complexes [36]. The contacts are grouped into charged-charged, charged-polar, charged-apolar, polar-polar, polar-apolar and apolar-apolar. In addition, the number of residue contacts per amino acid and the link density were explored. The number of residue contacts at the interface within an optimal distance cutoff of 5 Å was found to be a much better discriminant than the buried surface area since residue contacts also reflect the topology of the interface and not only its size. The number of apolar-apolar contacts showed the largest difference between the two classes of interfaces, followed by the number of aliphatic apolar residues such as leucine and valine. The properties of the non-interacting surface and the interaction energies, the latter being computationally expensive to compute, were excluded from the final predictor. Various machine learning algorithms were trained on the Many [27] dataset using the 22 best features, attaining an accuracy of 92% with a random forest classifier in a 10-fold cross validation, compared to 88% obtained by EPPIC. The methodology was recently implemented in the user-friendly webserver PRODIGY-CRYSTAL [37].

2.3.3. RPAIAnalyst (2018)

RPAIAnalyst [38] exploits correlated mutations of residue pairs across interface (RPAIs) by devising a new co-evolutionary feature. It performs direct coupling analysis and sharpens the resulting matrix using an image processing technique. Subsequently, the RPAIs that form intramolecular contacts in the monomer are eliminated, as are those separated by less than 5 positions in the sequence. The remaining 10 RPAIs with the highest scores are averaged in order to calculate the co-evolutionary

score for the interface. Notably higher scores were observed for biological interfaces with respect to crystal contacts. The random forest predictor also evaluates residue pair frequency, conservation score, Voronoi cell volume, secondary structure, core-rim, B-factor and hot spots. Hydrophobic-hydrophobic and hydrophobic-neutral RPAIs are more common in biological interfaces, while neutral-polar and polar-polar are mainly found in crystal contacts. The biological interfaces are also characterized by better conservation, smaller volumes, more residues in helical and less residues in coil conformation, as well as higher number of hot spots. The method considerably outperforms DiMoVo and PITA on the DC dataset when trained on their respective training datasets but shows only a slightly better performance than PISA in the same case. Training on the DC dataset [16] gives an accuracy (as reported in Equation (5)) of 84.6% in a 5-fold cross validation procedure, while EPPIC and Luo et al. arrive at 81% and 83.2%, respectively. However, due to the absence of large crystallographic interfaces in the DC dataset [16] the method appeared to have some difficulties detecting them.

### 2.3.4. NOXclass (2016)

NOXclass [22] discriminates between obligate, non-obligate and crystal packing interactions. It was one of the first methods to combine several interface properties using a machine learning algorithm. A two-stage support vector machine considers three properties: interface area, ratio of interface area to protein surface area and amino acid composition of the interface. Other features, such as correlation between amino acid compositions and protein surface, gap volume index and conservation score of the interface, were also investigated but were not found to substantially contribute to the prediction. The Support Vector Machine (SVM) was trained on a newly compiled dataset, Zhu et al. [22] (Table 2), consisting of 137 biological (75 obligate + 62 non-obligate) and 106 crystal packing interactions and achieved a first stage (distinguishing between biological and crystal interactions) accuracy of 97.9% in a leave-one-out cross-validation procedure. It was also tested on the Bahadur et al. dataset [15], on which it was less accurate (80%) in classifying crystal packing contacts due to their large interface areas.

### 2.3.5. IChemPIC (2015)

IChemPIC [39] relies on a majority vote of 10 independent random forest classifiers based on 45 features describing interface size, chemical complementarity and buriedness. The interface is represented using interaction pseudoatoms (IPAs). In addition to the total number of IPAs, the method also considers the percentage of each of the following interaction types: Hydrophobic contacts, hydrogen bonds, ionic bonds and aromatic interactions. Finally, for each of the four interaction types, the distribution of buriedness of the corresponding IPAs was binned into 10 intervals in the 25–100% range, resulting in a total of 40 auxiliary descriptors. A new dataset was compiled by merging the Bahadur et al. dataset [15] with the DC dataset [16] (Table 2). The resulting dataset was filtered and manually enriched, for a total of 200 biological and 200 crystallographic interfaces. The method was trained on the 75% of this dataset and tested on the remaining 25%. While biological interfaces have more IPAs than crystallographic ones, the percentages of interaction types are very similar between the two classes, with biological interfaces having slightly more hydrophobic contacts and slightly fewer aromatic interactions and hydrogen bonds. The number of IPAs and the percentage of fully buried hydrophobic contacts were revealed to be the most important features. On the test dataset, IChemPIC is 75% accurate, compared to the accuracy of 72%, 54%, 73% and 83% observed for NOXclass, DiMoVo, PISA and EPPIC, respectively. However, it is the only method among the five that showed balanced sensitivity (see Equation (1)) and specificity (see Equation (2)) values. While all methods performed very well on the Bahadur et al. [7] and Ponstingl et al. [23] dataset, IChemPIC was the second-best method.

### 2.3.6. Luo et al. (2014)

Another machine learning-based method was developed by Luo and colleagues [40]. Their random forest predictor uses 46 optimal features among which: Core-surface and core-interface scores, residue propensity, core area, non-polar area and fully buried atoms fractions, gap volume and local density indices, number of hot spots, interface area ratio, amino acid and secondary structure compositions, propensities of interface residues, core residues and hot spots. The analyses suggest that the biological interfaces are better packed with respect to the crystallographic ones, which is favorable for the formation of their larger core regions. Biological interfaces are also characterized by larger differences between their core and surface in terms of amino acid composition and evolutionary conservation. The method outperforms DiMoVo, PITA and PISA on the DC dataset [16] when trained on their respective training datasets. When trained on the DC dataset [16], it reaches an accuracy of 86.9% in a 5-fold cross validation, compared to 81% achieved by EPPIC.

### 2.3.7. IPAC (2011)

IPAC (Inference of Protein Assembly Crystals) [41] generates all symmetry-related molecules in a crystal and applies a naïve Bayes classifier to check the interfaces between them. Molecules that form biological interfaces are merged into functional units (FUs) and interfaces between the FUs are checked in an iterative procedure. Finally, the method performs several empirically identified Boolean checks and predicts the largest FU that satisfies point group symmetry. The classifier relies on the following ten features: Interface area, normalized surface complementarity, variation of accessible surface area, normalized interface packing, interface packing gradient, hydrophobicity of interface and surface, normalized surface complementarity and interface packing paired metric, patch ratio and normalized solvation energy. It was trained on a self-assembled dataset and on three different validation datasets showed a more stable performance (accuracy of 90%, 90% and 97%, respectively) to both DiMoVo (accuracy of 92%, 92% and 39%, respectively) and NOXclass (accuracy of 82%, 83% and 98%, respectively). On the second dataset it showed an assembly prediction accuracy of 90%, which is similar to or better than PITA and PISA, which reach 84% and 90%, respectively. With an accuracy of 95% compared to 53%, it significantly outperforms PISA on the third dataset.

### 2.3.8. DiMoVo (2008)

DiMoVo (DIscrimination between Multimers and MOnomers by VOronoi tessellation) [42] uses Voronoi tessellation of a coarse-grained protein structure where each residue is represented as a sphere. A total of 87 features were investigated for discriminating between biological and crystallographic dimers, but only 27 were retained in the final model. The method takes into consideration the interface area, the number of interface residues and their Voronoi volumes, the frequency of interface residues, the frequency of pairs of residues in contact (residues are grouped into 6 categories) and the distances between their geometric centers. The analysis shows that the interface area and the number of interface residues, even though largely correlated, are both very important for the prediction. The support vector machine reaches an accuracy of 95% on the training datasets by Bahadur et al. 2000 and 2003 [14,15] (Table 2) in a leave-one-out cross validation. When no prediction is made in highly uncertain cases, the accuracy increases by 1–2%, but the recall (see Equation (4)) drops, especially for the biological dimers.

**Table 1.** List of biological interface classifiers.

| Method | Methodology | Training Dataset | Webserver or Source Code | Ref. |
|---|---|---|---|---|
| **Methods with a functioning webserver or source code** | | | | |
| ClusPro-DC | Number of near-native docking poses in the top 1000 lowest energy structures | Bahadur et al. [14,15] | https://cluspro.bu.edu/dimer_predict | [12] |
| EPPIC | Number of core residues, core-rim and core-surface entropy ratios | DC [16], Many [27] [1] | http://www.eppic-web.org | [16,27,28] |
| IPAC | Interface area, normalized surface complementarity, accessible surface area variation, normalized interface packing, interface packing gradient, hydrophobicity of interface and surface, patch ratio and normalized solvation energy -> NB | Mitra and Pal [41] | http://pallab.serc.iisc.ernet.in/IPAC | [41] |
| PIACO | Covariation signal, number of core residues, amino acid compositions of the interface and of core residues, amino acid pair frequency, local density index, residue propensity score and gap volume index -> RF | DC [16] | https://github.com/yfukasawa/piaco | [33] |
| PISA | Binding energy and entropy of dissociation | Ponstingl et al. [19] | https://www.ebi.ac.uk/msd-srv/prot_int http://www.ccp4.ac.uk/pisa | [17,18,20] |
| PITA | Interface area and atom-pair frequencies | Ponstingl et al. [19,23] | https://www.ebi.ac.uk/thornton-srv/databases/pita | [19,23] |
| PRODIGY-CRYSTAL | Number of residue contacts grouped by their character, number of residue contacts per amino acid and link density -> RF | Many [27] | https://haddock.science.uu.nl/services/PRODIGY-CRYSTAL | [36,37] |
| RPAIAnalyst | Co-evolutionary and conservation scores, residue pair frequency, Voronoi cell volume, secondary structure, core-rim, B-factor and hot spots -> RF | DC [16] | http://liulab.hzau.edu.cn/RPAIAnalyst | [38] |

**Table 1.** *Cont.*

| Method | Methodology | Training Dataset | Webserver or Source Code | Ref. |
|---|---|---|---|---|
| **Methods without a functioning webserver or source code** | | | | |
| Bahadur et al. | Non-polar interface area, fraction of buried atoms and residue propensity scores | Bahadur et al. [14,15] | | [15] |
| CFPScore | Potential mean force and shape complementarity scores, interface size and packing density | Ponstingl et al. [23] | | [31] |
| DiMoVo | Interface area, number, Voronoi volumes and frequencies of interface residues, frequency of pairs of residues in contact and distances between their geometric centers -> SVM | Bahadur et al. [14,15] | http://fifi.ibbmc.u-psud.fr | [42] |
| Elcock and McCammon | Ratio between mean interface and surface entropies | | | [32] |
| IChemPIC | Number of interaction pseudoatoms, percentage of interaction pseudoatoms for each of the four interaction types and their corresponding distributions of buriedness -> RF | Da Silva et al. [39] | http://bioinfo-pharma.u-strasbg.fr/IChemPIC | [39] |
| Liu et al. | Average B-factor score | DC [16] | | [21] |
| Luo et al. | Core-surface and core-interface scores, residue propensity, core area, non-polar area and fully buried atoms fractions, gap volume and local density indices, number of hot spots, interface area ratio, amino acid and secondary structure compositions and propensities of interface residues, core residues and hot spots -> RF | DC [16] | http://cic.scu.edu.cn/bioinformatics/bio-cry.zip | [40] |
| NOXclass | Interface area, ratio of interface area to protein surface area and amino acid composition of the interface -> SVM | Zhu et al. [22] | http://noxclass.bioinf.mpiinf.mpg.de | [22] |
| PreBI and COMP | Interface area and shape, hydrophobicity andelectrostatic potential | Tsuchiya et al. [29] | http://pre-s.protein.osaka-u.ac.jp/prebi | [29,30] |
| Valdar and Thornton | Fraction of buried surface residues and contact conservation score -> MLP | Ponstingl et al. [23] | | [43] |

[1] Linear regression classifier that outputs the probability for an interface to be biological in EPPIC 3 [28] was trained on this dataset.

### 2.3.9. Valdar and Thornton (2001)

Valdar and Thornton [43] were among the first to investigate the role of size and conservation in classifying interfaces. Starting from the Ponstingl et al. dataset [23], they found that the biological interfaces are generally larger and better conserved than non-biological contacts. However, biological contacts were not always highly conserved and well conserved contacts were not solely biological. Biases in the multiple sequence alignment were proposed as a possible explanation in the former case, while in the latter the presence of binding sites was sometimes the cause. In the absolute assessment of homodimeric contacts, a multilayer perceptron relying on both size and conservation showed a comparable performance to a linear heuristic discriminator, while the latter was slightly superior in its relative assessment.

## 3. Datasets

Reliable datasets of biological and crystallographic interfaces are essential for the training of new predictors and their comparison. Some of the first methods were developed on the dataset compiled in 2000 by Ponstingl and coworkers [23]. Subsequently, they assembled another dataset consisting of 163 oligomers and 55 monomers on which PITA and PISA were trained. Bahadur et al. [14,15] created two other popular datasets, containing 122 homodimers and 188 monomers with interfaces of at least 800 $\text{Å}^2$, respectively. These two are often combined, as was done for the development of DiMoVo [42] and ClusPro-DC [12]. In order to train NOXclass [22], Zhu and colleagues put together a dataset of obligate, non-obligate and crystallographic interfaces, while Mitra and Pal used the PiQSi database [44] for generating a larger dataset on which IPAC [41] was trained. The manually annotated DC dataset [16], composed of 83 biological and 82 crystallographic interfaces, specifically concentrates on the range of interface sizes between 1000 and 2000 $\text{Å}^2$, because these are more difficult to predict. In addition to for developing EPPIC, it was used in the development of other methods such as those by Liu et al. [21] and Luo et al. [40], but also RPAIAnalyst [38] and PIACO [33]. De Silva et al. combined it with the dataset by Bahadur et al. [15] in order to train IChemPIC. A large-scale dataset called Many [27] was automatically created based on the ProtCID database [10]. It consists of almost 5800 entries that were exploited in the development of PRODIGY-CRYSTAL [36].

**Table 2.** List of biological interface datasets.

| Dataset | Content | Ref. |
|---|---|---|
| Bahadur et al. | 122 homodimers and 188 monomers | [14,15] |
| DC | 83 biological and 82 crystallographic interfaces | [16] |
| De Silva et al. | 200 biological and 200 crystallographic interfaces | [39] |
| Many | 2831 biological and 2913 crystallographic interfaces | [27] |
| Mitra and Pal | 268 dimers and 396 monomers | [41] |
| Ponstingl et al. (2000) | 76 homodimers and 96 monomers | [23] |
| Ponstingl et al. (2003) | 163 oligomers and 55 monomers | [19] |
| Zhu et al. | 137 biological (75 obligate + 62 non-obligate) and 106 crystallographic interfaces | [22] |

## 4. Conclusions

Over the past decades, many efforts have been devoted to the extensive study and characterization of protein interfaces. The increasing number of structures of protein complexes determined through X-ray crystallography has allowed a good characterization of crystallographic interfaces, which are quite different from the biological ones in terms of amino acids composition and stability. This has given rise to computational methods that, while employing such properties, can classify protein interfaces as biological or crystallographic, reaching good coverage and performance.

While there is no single property that can, on its own, completely discriminate between the two types of interfaces, a combination of those can efficiently distinguish between interfaces (often in more than 90% of the cases). Besides methods based on evolution or stability (energy) evaluation, recent

years have seen the rise of machine learning-based approaches, facilitated by the increasing availability of training datasets.

Besides their application to interface classification in crystallographic structures, these methodologies have an application potential beyond this specific field. A clear link exists, for example, between this problem and the field of docking. The two problems are highly similar: In both cases, one should distinguish the interface which plays a biological role from the others, which are either artefacts of the crystallization process, in the case of crystallographic interface classification, or of in silico simulations, in the case of docking. For example, an application of docking applied to the protein interface classification problem has already been implemented in ClusPro-DC [12]. More exchanges between those two fields can be expected in the future.

**Author Contributions:** Review and editing, K.E., A.V. and A.M.J.J.B.; visualization, K.E. and A.V.; supervision, A.V. and A.M.J.J.B.; project administration, A.V.; funding acquisition, A.M.J.J.B. All authors have read and agreed to the published version of the manuscript.

**Funding:** This research received no external funding.

**Conflicts of Interest:** The authors declare no conflict of interest.

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
