# Peer review of "Biological vs. Crystallographic Protein Interfaces: An Overview of Computational Approaches for Their Classification"

_crystals, doi:10.3390/cryst10020114_

Round 1
Reviewer 1 Report
The authors review the field of computational crystal interface classification. They do a thorough job on describing the many proposed methods in literature, reporting some of the goods and bads of each of them.
Comments
Page 3, lines 91, 92: it would be good to say what Positive and Negative mean in this context. Also some authors have used recall and other synonyms to this terms, it would be nice to see those described here too. Of course elsewhere in the paper it is good to stick to the ones you choose for consistency. Some other measures are also present in literature (e.g. Matthews Correlation Coefficient or F1), it’d be good to describe those too for completeness.
Also a minor point about the equations, please remove the commas between them. They make things confusing, for a moment I thought they were part of the equations.
Section 2: a comment on the 3 categories used (Energy-based, Empirical/KB-based and ML-based). The authors place PISA and ClusPro-DC in the energy-based category. However what the methods do is merely approximating the energy by fitting many empirical parameters to the training sets. I would suggest that the authors clarify that point in the manuscript.
In the description of existing methods (Section 2) there’s no mention of methods that make use of the PDB as a knowledge base for inferring biological relevance of interfaces/assemblies. Good examples are: ProtCID (Xu et al 2010), QSBio (Dey et al 2018).
I would also like to suggest that authors include a citation to the review on the same topic by Capitani et al, Bioinformatics 2016.
Author Response
Reviewer: The authors review the field of computational crystal interface classification. They do a thorough job on describing the many proposed methods in literature, reporting some of the goods and bads of each of them.
Page 3, lines 91, 92: it would be good to say what Positive and Negative mean in this context. Also some authors have used recall and other synonyms to this terms, it would be nice to see those described here too. Of course elsewhere in the paper it is good to stick to the ones you choose for consistency. Some other measures are also present in literature (e.g. Matthews Correlation Coefficient or F1), it’d be good to describe those too for completeness.
Also a minor point about the equations, please remove the commas between them. They make things confusing, for a moment I thought they were part of the equations.
Response: We wish to thank the Reviewer for those helpful comments that will improve the readability of our manuscript. We gladly introduced the suggestions reported (see page 4 for the “Recall” formula, and lines 103-104: “In this context, the positive class is defined as biological, while the negative class is that of the crystallographic interfaces”).
Regarding the description of other possible measures, the statistical criteria used for performance assessment are many and we believe that reporting an exhaustive description of those not mentioned in our work is out of the scope of the reviewer.
Reviewer: Section 2: a comment on the 3 categories used (Energy-based, Empirical/KB-based and ML-based). The authors place PISA and ClusPro-DC in the energy-based category. However what the methods do is merely approximating the energy by fitting many empirical parameters to the training sets. I would suggest that the authors clarify that point in the manuscript.
Response: We clarified this point, see page 4 lines 93-96: “Energy-based techniques determine the stability of an interface using a scoring function which approximates the energy of a complex. The methods in the other two groups consider different properties of the interface, either by deriving heuristic rules (empirical knowledge-based), or by using specific algorithms (machine learning-based).”
Reviewer: In the description of existing methods (Section 2) there’s no mention of methods that make use of the PDB as a knowledge base for inferring biological relevance of interfaces/assemblies. Good examples are: ProtCID (Xu et al 2010), QSBio (Dey et al 2018).
Response: The reason we excluded such comparative methods from our work is because they do not operate on isolated crystal structures, but rather on a cluster of structures from the PDB. For completeness, we now briefly mention them, see page 4 lines 97-99: “Finally, approaches that rely on a comparison between interfaces of homologous proteins across multiple crystal structures also exist, such as ProtCID [10] and QSbio [11], but they were not the focus of this work.”
Reviewer: I would also like to suggest that authors include a citation to the review on the same topic by Capitani et al, Bioinformatics 2016.
Response: The review is definitely of great interest for the reader, we thank the Reviewer for pointing it out. We now added the information page 3 lines 81-82: “For other valuable reviews in the field, please refer to those by Capitani et al. [4] and Xu et al. [9].”
Reviewer 2 Report
The manuscript by Elez et al. is a thorough review of the different computational methods available for the identification of biological and crystallographic protein interfaces. The manuscript is clear and well written. My only suggestion is to provide either via table or in the conclusion section some of the differences between biological and crystallographic interfaces reached comparing all the methods. Suggest a methodology to follow when one is doing such analysis.
Author Response
Response: We wish to thank the Reviewer for kind comments regarding our work. We have chosen to write a review on prediction models in order to provide a practical overview of the computational approaches used (energy-based, machine learning and so on), rather than a comparison of the methods. Many methods show very different implementation, training and validation approaches, which would make difficult to compare them assuring complete fairness.
Reviewer 3 Report
The manuscript represents a reference of methods developed for the discrimination between “biological” and “crystal” interfaces, which are supposed to help identification of the former from crystal packing. This problem is indeed important for structural biologists, trying to infer on molecular interactions from the results of crystallographic studies. The manuscript is written in clear language, and the reference of various methods is fairly complete. I also do not have any reservations in regard to the factual content, and find that, to my knowledge, all methods are described correctly and in a manner sufficient for understanding and general orientation.
Although I think that the manuscript can be published as is for there is nothing wrong in it and the content is generally useful, its value could be considerably increased if authors have provided a more extensive discussion of the problem and results of their overview. For example, the definition of the “biological” interface is not given in any certain terms — and this is a relatively murky side of any research in this field, which is a problem by itself. What authors offer as such a definition, is actually half of sentence in line 53: “The biologically relevant one, corresponding to the interface occurring in solution and eliciting the biological function”, and that is rather vague and incomplete. For example, interfaces in a stable complex, which never dissociates, do not elicit any biological function. In this case, it is _inter_, rather than _intra_, -complex interfaces that are “biologically” relevant, in full contradiction to what is obtained with methods aimed at finding most persistent interactions in crystal packing. The authors could also mention transient interactions, which are very important for many processes in the cell, but can be hardly identified as “occurring in solution”, because, in transient sense, any interface may occur in solution for a fraction of millisecond. Such interactions can be rarely found in crystals. “Occurrence” (which authors, probably, equivalence with a relative persistence in chemical equilibrium) in solution is also subject to chemical conditions and general environment, so that the same interaction or interface may or may not be detected or work as assumed depending on circumstances (such as, protein concentration, pH, salinity, etc). There are classes of interactions, such as FAB, which are particularly difficult for automatic identification, and review could have highlighted this and similar facts, possibly prompting a research in particular directions. The very assumption that a particular “biological” interface is found in crystal packing as it would be presented in solution, is merely a hypothesis, and, possibly, the review should have started from this statement. There is a lot of what can be said about macromolecular interactions and interfaces in a review of methods developed to detect them. It may well appear then that different methods are developed with different understanding of what a “biological” interface is, and, therefore, they should not be compared with each other or taken as alternatives. This list of suggestions is, of course, far from being complete.
Author Response
Reviewer: The manuscript represents a reference of methods developed for the discrimination between “biological” and “crystal” interfaces, which are supposed to help identification of the former from crystal packing. This problem is indeed important for structural biologists, trying to infer on molecular interactions from the results of crystallographic studies. The manuscript is written in clear language, and the reference of various methods is fairly complete. I also do not have any reservations in regard to the factual content, and find that, to my knowledge, all methods are described correctly and in a manner sufficient for understanding and general orientation.
Response: We thank the Reviewer for kind comments regarding our work.
Reviewer: Although I think that the manuscript can be published as is for there is nothing wrong in it and the content is generally useful, its value could be considerably increased if authors have provided a more extensive discussion of the problem and results of their overview. For example, the definition of the “biological” interface is not given in any certain terms — and this is a relatively murky side of any research in this field, which is a problem by itself. What authors offer as such a definition, is actually half of sentence in line 53: “The biologically relevant one, corresponding to the interface occurring in solution and eliciting the biological function”, and that is rather vague and incomplete. For example, interfaces in a stable complex, which never dissociates, do not elicit any biological function. In this case, it is _inter_, rather than _intra_, -complex interfaces that are “biologically” relevant, in full contradiction to what is obtained with methods aimed at finding most persistent interactions in crystal packing.
The authors could also mention transient interactions, which are very important for many processes in the cell, but can be hardly identified as “occurring in solution”, because, in transient sense, any interface may occur in solution for a fraction of millisecond. Such interactions can be rarely found in crystals. “Occurrence” (which authors, probably, equivalence with a relative persistence in chemical equilibrium) in solution is also subject to chemical conditions and general environment, so that the same interaction or interface may or may not be detected or work as assumed depending on circumstances (such as, protein concentration, pH, salinity, etc). There are classes of interactions, such as FAB, which are particularly difficult for automatic identification, and review could have highlighted this and similar facts, possibly prompting a research in particular directions. The very assumption that a particular “biological” interface is found in crystal packing as it would be presented in solution, is merely a hypothesis, and, possibly, the review should have started from this statement. There is a lot of what can be said about macromolecular interactions and interfaces in a review of methods developed to detect them.
Response: We have now extended the definition of interfaces in order to facilitate its comprehension, see page 2 lines 54-55: “Specifically, assemblies having a Kd value in the low micromolar range or smaller are usually considered biological, unlike weak interactions which are characterized by higher Kd values [3,4].”.
We agree with the Reviewer that the general topic of macromolecular interactions is broad and complex. In order to address it fully, a much more comprehensive discussion would be needed, which is out of the scope of this work. Besides providing an introduction to the problem of interface classification, we have now added a few reviews in the field for the readers that wish to explore this part more (see page 2 lines 56-57: “Further information regarding the extensive topic of protein interactions can be found in dedicated reviews, such as those by Nooren and Thornton [5] or Marsh and Teichmann [6].”)
Reviewer: It may well appear then that different methods are developed with different understanding of what a “biological” interface is, and, therefore, they should not be compared with each other or taken as alternatives. This list of suggestions is, of course, far from being complete.
Response: Considering the definitions given by our colleagues in the field, we believe that there is a common understanding between all of us and that we indeed face the same challenges. We cite here below the definition provided in some of the most cited articles in the field:
Liu et al. (2006) – “The known protein–protein complexes can be termed as biological complex, because they are known to associate in solution. Most crystal contacts are artifacts of crystallization that would not occur in solution, which are termed as non-biological contacts.”
Zhu et al. (2006) – “However, not all interactions observed in structures of protein complexes determined by X-ray crystallography are biologically relevant. Many of them are formed during the crystallization process and would not appear in vivo. Such crystal packing contacts are non-specific and have no biological function associated [2].”
Duarte et al. (2012) – “Protein crystal lattices contain two kinds of interfaces: biological ones (as present in physiological conditions) and crystal packing ones (non-specific), indistinguishable by crystallographic means.”
Da Silva (2015) – “..., we will consider as biological any protein−protein complex with a true biological relevance and function (e.g., cell adhesion, cell signaling, immune recognition, transcription). Homo- or hetero-oligomeric complexes resulting either from crystal packing or lacking any known biological function will be considered crystallographic.”